# The Anti-Proliferative Activity of Secondary Metabolite from the Marine *Streptomyces* sp. against Prostate Cancer Cells

**DOI:** 10.3390/life11121414

**Published:** 2021-12-16

**Authors:** Hung-Yu Lin, Yong-Shiou Lin, Shou-Ping Shih, Sung-Bau Lee, Mohamed El-Shazly, Ken-Ming Chang, Yu-Chen S. H. Yang, Yi-Lun Lee, Mei-Chin Lu

**Affiliations:** 1School of Medicine, College of Medicine, I-SHOU University, Division of Urology, Department of Surgery, E-Da Cancer & E-Da Hospital, Kaohsiung 824, Taiwan; ed100464@edah.org.tw; 2Graduate Institute of Marine Biology, National Dong Hwa University, Pingtung 944, Taiwan; 398157@cch.org.tw; 3Vascular and Genomic Center, Institute of ATP, Changhua Christian Hospital, Changhua 500, Taiwan; 4Doctoral Degree Program in Marine Biotechnology, National Sun Yat-Sen University, Kaohsiung 804, Taiwan; d065620003@nsysu.edu.tw; 5Doctoral Degree Program in Marine Biotechnology, Academia Sinica, Taipei 115, Taiwan; 6Ph.D. Program in Drug Discovery and Development Industry, College of Pharmacy, Taipei Medical University, Taipei 115, Taiwan; sbl@tmu.edu.tw; 7Department of Pharmacognosy, Faculty of Pharmacy, Ain-Shams University, Organization of African Unity Street, Cairo 115, Egypt; mohamed.elshazly@pharma.asu.edu.eg; 8Department of Pharmaceutical Biology, Faculty of Pharmacy and Biotechnology, German University in Cairo, Cairo 118, Egypt; 9Department of Pharmacy, Tajen University, Pingtung 907, Taiwan; 1600216@hcch.org.tw; 10Department of Pharmacy, Hengchuen Christian Hospital, Pingtung 946, Taiwan; 11Joint Biobank, Office of Human Research, Taipei Medical University, Taipei 110, Taiwan; 12Department of Urology, Pingtung Hospital, Ministry of Health and Welfare, Pingtung 944, Taiwan; 13National Museum of Marine Biology & Aquarium, Pingtung 944, Taiwan

**Keywords:** Lu01-M, marine actinomycetes, *Streptomyces* sp., prostate cancer

## Abstract

Many active substances from marine organisms are produced by symbiotic microorganisms such as bacteria, fungi, and algae. Secondary metabolites from marine actinomycetes exhibited several biological activities and provided interesting drug leads. This study reported the isolation of Lu01-M, a secondary metabolite from the marine actinomycetes *Streptomyces* sp., with potent anti-proliferative activity against prostate cancers. Lu01-M blocked cell proliferation with IC_50_ values of 1.03 ± 0.31, 2.12 ± 0.38, 1.27 ± 0.25 μg/mL in human prostate cancer PC3, DU145, and LNCaP cells, respectively. Lu01-M induced cytotoxic activity through multiple mechanisms including cell apoptosis, necroptosis, autophagy, ER stress, and inhibiting colony formation and cell migration. Lu01-M induced cell cycle arrest at the G2/M phase and DNA damage. However, the activity of autophagy induced survival response in cancer cells. Our findings suggested that Lu01-M holds the potential to be developed as an anti-cancer agent against prostate cancers.

## 1. Introduction

The study of marine natural products gained momentum in the last few decades with the introduction of therapeutic drugs of marine origin. Marine natural products not only possess structural diversity and potent biological activities but also exhibit favorable safety profiles [1]. Coastal Asian countries are blessed with a diverse marine environment. Taiwan is surrounded by one of the highest coral densities and diversities in the world. Such an environment led to significant advancements in research and technologies related to the marine environment [2,3]. Special emphasis was put on the discovery of new compounds with potent biological activities, especially anticancer agents. Not only marine organisms were subjected to intensive investigation but also their associated symbiotic organisms such as bacteria, fungi, and algae that proved to be the source of interesting active compounds [4]. Okadaic acid was isolated from the sponges *Halihondria okadai*, *H. melanodocia*, and *Prorocentrum* sp. This compound inhibited specific protein phosphatase and exhibited potent cytotoxic activity [5]. The secondary metabolite “Safracin B” isolated from *Pseudomonas fluorescens* sp. showed similarity to Ecteinascidin-743 (ET-743), which was isolated from *Ecteinascidia turbinata*. Safracin B was easily converted to ET-743 via chemical modification [6]. These results indicated that microorganisms may be involved in the metabolism of natural products of marine organisms.

Cancer therapeutics act through different mechanisms, including anti-proliferative activity or inducing the death of tumor cells. Cell cycle arrest is a common mechanism to inhibit cell growth. Cell cycle arrest is usually initiated by DNA damage or DNA repair inhibition [7]. There are several types of cell death, including apoptosis, necrosis, and autophagy [8]. Apoptosis is proceeded through major mechanisms including intrinsic and extrinsic pathways. The intrinsic pathway of apoptosis, also known as the mitochondrial pathway, is initiated by the cell itself in response to damage. The extrinsic pathway, also known as the death receptor pathway, is induced via death receptors stimulated by the cells of the immune system [8]. Necrosis is an uncontrolled form of cell death that is caused by an external damaging factor such as hypoxia or inflammation. Necroptosis is another type of cell death with similar characteristics to necrosis but with tightly regulated signaling pathways [8].

Prostate cancer (PC) is the second most common cancer type in men with 1.4 million new cases in 2020, and it is often developed above 50 years old [9]. Androgen deprivation therapy is the major therapy for recurrent prostate cancer. However, these patients will eventually develop resistance and become castration-resistant prostate cancer (CRPC). In the currently approved treatment protocols, docetaxel was combined with prednisone and resulted in prolongation of survival for 2.8 months [10], sipuleucel-T resulted in 4.1 months [11], abiraterone acetate led to an additional 4 months [12], and enzalutamide prolonged the lifetime for 4.8 months [13] in metastatic-CRPC. However, these treatments suffer from certain drawbacks and did not show a significant improvement in the patient’s life span of more than few months.

In the present study, we isolated novel metabolites Lu01-M from bacterial marine sediments. Lu01-M showed cytotoxic activity via cell apoptosis, inhibition of colony formation, suppression of cell migration, cell cycle arrest, promotion of DNA damage, necroptosis, and stimulation of autophagy in prostate cancer cell lines. These results suggested that Lu-01 has the potential to be developed as an anti-cancer drug agent against prostate cancer.

## 2. Materials and Methods

### 2.1. Cell Culture and Chemicals

Dimethylsulfoxide (DMSO), 3-(4,5-dimethylthylthiazol-2-yl)-2,5-diphenyl-tetrazolium bromide (MTT), and all other chemicals were purchased form Sigma-Aldrich (St. Louis, MO, USA). Antibodies against PARP, caspase-3, caspase-8, caspase-9, Bak, IRE1α, PERK, TNFR2, CDK6, p-cdc2, p-Rb (ser795), p-ATM, p-ATR, p-Chk1, and p-Chk2 were purchased from Cell Signaling Technologies (Beverly, MA, USA). Antibodies of Bcl-2, LC3, NF-κB, p-H2AX, ATF6, and CDK4 were purchased from Santa Cruz Biotechnology (Santa Cruz, CA, USA). Antibodies of TNFR1, RIPK1, RIPK3, and MLKL were purchased from ABclonal (ABclonal, Wuhan, China). Antibodies of Actin and p62/SQSTM1 were purchased from Sigma (Sigma, CA, USA). Anti-mouse and rabbit IgG peroxidase-conjugated secondary antibody were purchased from Pierce (Rockford, IL, USA). Hybond ECL transfer membrane and ECL Western blotting detection kits were purchased from Ameisham Life Sciences (Amersham, Buckinghamshire, UK). Rhodamine 123 and the carboxy derivative of fluorescein (carboxy-H2DCFDA) were purchased from Molecular Probes and Invitrogen detection technologies (Carlsbad, CA, USA).

### 2.2. 16S rDNA Sequencing

The Manual of Geno Plus Genomic DNA Extraction Midiprep System (Viogene-BioTek, Taipei, Taiwan) was followed for DNA purification from bacteria. The 16S rDNA were amplified by PCR with forwarding primer: 8F (5’-AGAGTTTGATCG-TGGCTCAG-3’) and reverse-primer: 1492R (5’-TACGGYTACCTTGTTACGAC-TT-3’). The PCR products were blasted by NCBI (The National Center of Biotechnology Information database).

### 2.3. Cell Culture

Human prostate cancer cell lines PC3, LNCaP, and DU145 were purchased from ATCC. PC3 was cultured in DMEM/F12 medium, LNCaP was cultured in RPMI 1640 medium, and DU145 was cultured in MEM medium. They all contained 10% FBS, antibiotics, and 1 mM sodium pyruvate.

### 2.4. MTT Assay

Cells (7 × 10^4^ cells/well) were seeded in a 96-well plate. After the cells were attached, Lu01-M was added at concentrations of 0.78, 1.56, 3.125, 6.25, and 12.5 μg/mL. DMSO was used as the solution control. The cells were incubated for 24, 48, and 72 h at 37 °C and 5% CO_2_. After adding 50 μL of a solution mixed with 500 mL PBS and 1 g MTT (thiazolyl blue tetrazolium bromide, Sigma-M2128), cells were incubated for 1–4 h and centrifuged at 2000 rpm for 2 min. Then, the supernatant was removed, DMSO (200 μL) was added, and it was placed in an 80 rpm shaker and shaken until the purple crystals were completely dissolved. The absorbance was measured at 570 nm and 620 nm with an ELISA Reader (Bio-Rad, Hercules, CA, USA). Then, we subtracted the values obtained from OD570 nm and OD620 nm and used the OD value of the DMSO group (control group) as the 100% cell viability rate. Then, we compared the OD values obtained under different drug concentration treatments with the OD values of the DMSO group to obtain the cell viability rates, respectively. Then, we entered the cell survival rate corresponding to different concentrations through the CalcuSyn software (Bio-soft, Ferguson, MO, USA) and calculated the IC_50_ value by interpolation.

### 2.5. Colony Formation Assay

Cells (5 × 10^2^ cells/well) were seeded in a 6-well plate. After the cells were attached, Lu01-M was added at concentrations of 0.78, 1.56, 3.125, 6.25, and 12.5 μg/mL. DMSO was used as the control. After 6 h of incubation, a fresh culture medium with no drugs replaced the old culture medium. The culture medium was changed every four days. After culturing for twelve days, PBS was used for wash, the cells were fixed with 2% paraformaldehyde for 30 min; then, they were stained with crystal violet (0.4 g/L) for 10 min, washed twice with PBS, and then dried with distilled water. The stained communities were counted.

### 2.6. Wound Healing Assay

Cells (8 × 10^5^ cells/well) were seeded in a 12-well plate. After the cells were attached, a sterilized 200 μL plastic tip was used to scrape a wound of equal width on the bottom of each well. The culture medium was removed and washed once with 1 mL of PBS to wash off the scraped cells. The culture medium was replaced, and Lu01-M was added at different concentrations: 1.56, 3.125, and 6.25 μg/mL at the same time. DMSO was used as the control. The disc was placed under the microscope, and the image was recorded at 10× magnification. A digital camera (Nikon P5100) was used to take pictures of the wound width at a fixed position. The pictures were taken every 3 h within 12 h. Hour zero was used as the reference point to quantify the cell gap area into the graph.

### 2.7. Western Blotting

PC3 cells (7 × 10^5^ cells/mL) were seeded in a 10 cm culture dish and were treated with 1.56, 3.125, and 6.25 μg/mL of Lu01-M. The cells were cultured at 37 °C and 5% CO_2_ in a cell incubator for 24 h. After treatment with Lu01-M (6.25 μg/mL) for 6, 12, and 24 h, the cells were collected and were lysed by RIPA lysis buffer (including protease inhibitor, NaF, and Na_3_VO_4_). SDS-polyacrylamide (10%, 12%, or 15%) was loaded for electrophoresis separation and transferred to the PVDF membrane. The PVDF membrane was immersed with the specific primary antibody and secondary antibody. The substrate ECL luminescence coloring system was used to detect the expression of specific proteins.

### 2.8. Cell Cycle

Cells (1 × 10^5^) were seeded in a 6-well plate overnight. The culture medium was replaced without FBS for 24 h of treatment. The culture medium containing FBS was replaced, and Lu01-M (1.56, 3.125, and 6.25 μg/mL) was added. DMSO was used as the control. The cells were incubated at a 37 °C and 5% CO_2_ cell for 24 h. The cells were trypsinized and were fixed with 70% alcohol. The cells were resuspended in 1× PBS, 0.2% Triton-100 was added to break the cell membrane, and RNase A was added at the same time. PI was added reaching the final concentration of 40 μg/mL to be embedded in the DNA, and the DNA content was analyzed by flow cytometry (Becton-Dickinson, San Jose, CA, USA). Finally, the MultiCycle AV software (Phoenix Flow Systems, San Diego, CA, USA) was used to analyze the cell cycle distribution.

### 2.9. Mitochondrial Membrane Potential

Cells (2 × 10^5^ cells/well) were seeded in a 6-well plate. After the cells were attached, Lu01-M was added (1.56, 3.125, and 6.25 µg/mL). DMSO was used as the control. The cells were incubated at 37 °C and 5% CO_2_ cell for 24 h. The cells were stained with Rhodamine 123 and were detected by flow cytometry (Becton-Dickinson, San Jose, CA, USA).

### 2.10. Immunofluorescence Assay

The cells (2 × 10^5^ cells) were cultured in a 12-well plate with the slides at the bottom. After the cells were attached, Lu01-M (1.56, 3.125, and 6.25 μg/mL) was added for 24 h. DMSO was used as the control. The medium was removed, and 2% paraformaldehyde was added to fix the cells for 30 min. The cells were washed three times with PBS, and then methanol was added. The cells were left to stand for 3 min and were washed three times with PBS. The cells were treated with a blocking buffer (1% BSA + 5% normal goat serum + 0.3% Triton in PBS) for 30 min at room temperature and were washed once with PBS. The appropriate amount of the primary antibody was dissolved in the blocking buffer (without Triton); then, they were separately added and were left at 4 °C overnight. The cells were washed three times with PBS on the second day. The fluorescent secondary antibody was prepared and was left at room temperature for 1 h. The cells were washed three times with PBS, and finally, DAPI was added at a concentration of 400 μg/mL for 5 min. The cells were washed once with PBS and twice with DDW. The slides were mounted with antifade reagent, and the cell morphology was observed with a fluorescent microscope (Olympus, Tokyo, Japan).

### 2.11. Statistics

The results were expressed as mean ± standard deviation (SD). Each experiment was performed using an unpaired Student’s *t*-test. A *p*-value of less than 0.05 was considered to be statistically significant.

## 3. Results

### 3.1. Separation of Bacteria and Metabolites with Antibacterial Activity from Marine Sediments

The marine sediments were collected at a depth of 400 m, Kaohsiung, Taiwan. The marine sediment samples were frozen for five days, and some microorganisms were physically removed before the culture. Two antibiotics, nalidixic acid and nystatin, were used to inhibit the growth of fungi and other bacteria. To distinguish whether the microorganisms obtained from the marine sediment samples are of marine origin or contaminated by terrestrial microorganisms, M0 and M1 media were used to test their seawater demand. Among the strains remaining after selection, there were six marine actinomycete strains with antibacterial activity by screening against *Staphylococcus aureus* and *Escherichia coli* (Table 1). Among the tested strains, four strains with higher activity were identified by Sanger sequencing the 16S rRNA region (Table 2). The metabolites were extracted from the fermentation broth. The fermentation broth of these four strains was extracted with ethyl acetate (EA) and was extracted with methanol (MeOH) to obtain the crude extract. The process is shown in Figure 1. In the cytotoxicity assay, only the metabolite Lu01-M extracted from Lu01 showed cytotoxic activity against PC3 cell lines.

### 3.2. Lu01-M Exhibited Anti-Proliferative Activity and Induced Cell Apoptosis in Prostate Cancer Cell Lines

Three prostate cancer cell lines, PC3, DU145, and LNCaP, were used in our experiments to evaluate the cytotoxic activity of Lu01-M. It inhibited PC3, DU145, and LNCaP cell proliferation with IC_50_ 1.03 ± 0.31, 2.12 ± 0.38, and 1.27 ± 0.25 μg/mL, respectively after 72 h of treatment (Figure 2a). The results indicated that PC3 cells were the most sensitive cancer cell line, and the cytotoxic activity was demonstrated in a dose- and time-dependent manner. Alternatively, the effect of Lu01-M on the cytotoxicity of normal CCD-966-SK cells (IC_50_: 1.56 μg/mL) is less toxic than the effect of the clinical anticancer drug, Doxorubicin (IC_50_: 0.0574 μg/mL) for 72 h treatment with MTT assay. Then, the apoptotic markers, sub-G1 and PARP cleavage, in prostate cancer cells PC3 increased with Lu01-M treatment after 72 h (Figure 2b,c). The percentage of sub-G1 increased from 3.14%, 5.3725%, and 6.995% to 29.6625% with the increasing concentration of Lu01-M (0, 1.56, 3.125, and 6.25 μg/mL). The results suggested that Lu01-M exhibited potent cytotoxic activity.

### 3.3. Lu01-M Inhibited Colony Formation and Cell Migration in Prostate Cancer PC3 Cell Lines

Lu01-M treatment inhibited the survival rate of prostate cancer PC3 cells (Figure 1). Therefore, we explored whether the use of different concentrations and long-term treatment can effectively inhibit the growth of cancer cells using colony formation assay. The results showed that the number of colonies formed by PC3 cancer cells decreased with the increase in drug concentration (Figure 3a). The use of Lu01-M (1.56 μg/mL) showed a significant reduction in the formation of cancer cell colonies (Figure 3b). The results confirmed that Lu01-M inhibited prostate cancer PC3 cells’ growth. Lu01-M affected the cell migration ability of PC3 cells as demonstrated by the wound-healing assay (Figure 3c). The treatment of PC3 cells with Lu01-M (3.125 and 6.25 μg/mL) significantly blocked the migration ability (Figure 3d).

### 3.4. Lu01-M Induced Cell Cycle G2/M Phase Arrest in PC3 Prostate Cancer Cell Lines

PC3 cells were treated with Lu01-M (0, 1.56, 3.125, and 6.25 µg/mL) for 24 h, and the cell cycle population was detected by flow cytometry. Lu01-M (3.125 and 6.25 µg/mL) caused cell cycle G2/M phase accumulation (Figure 4a,b). The expression of the cell cycle-related proteins CDK4, CDK6, p-cdc2, CyclinB1, p-Rb, and E2F1 was decreased as observed from the results of the Western blotting analysis (Figure 4c). The tumor suppressor gene p21 was significantly increased with Lu01-M treatment (Figure 4c).

### 3.5. Lu01-M Induced Mitochondrial Dysfunction and ER Stress in PC3 Prostate Cancer Cell Lines

Mitochondria contribute to many cellular functions and dysfunction including regulated calcium release, cell proliferation and differentiation, cell cycle, and cell death. The disruption of mitochondrial membrane potential (MMP) was determined by flow cytometry. MMP was significantly decreased with Lu01-M treatment in PC3 cells (Figure 5a). It was reduced from 4.2%, 11.4%, and 35.2% to 85.2%. Additionally, cytochrome *c* release from mitochondrial was also observed (Figure 5c). Western blot analysis of the expression ratio of cytochrome c/actin after treatment with different concentrations of Lu01-M showed that treatment and control increased significantly in a dose-dependent manner (Figure 5d). On the other hand, Lu01-M also caused ER stress (Figure 5e). According to the results of Western blotting assay, the expression of ER stress-related proteins including PERK, IRE1, Grp78/Bip, p-eIF-2α, ASK-1, and p-JNK increased after Lu01-M treatment (6.25 μg/mL) for 6, 12, and 24 h (Figure 5e), but the expression of ATF6 decreased (Figure 5e).

### 3.6. Lu01-M Caused DNA Damage in Human Prostate Cancer PC3 Cells

Lu01-M caused DNA damage in PC3 cells. The Western blotting results showed that the Lu01-M treatment significantly changed the phosphorylation of DNA damage target proteins ATR, ATM, Chk2, H2AX, and BRCA1 (Figure 6). Lu01-M treatment promoted the expression of p-ATM, p-Chk2, and p-H2AX proteins, but the expression of p-ATR, p-Chk1, and p-BRCA1 proteins was decreased (Figure 6).

### 3.7. Lu01-M Caused Necroptosis in Human Prostate Cancer PC3 Cells

Necroptosis was proved to be an independent mechanism of cell death from apoptosis or autophagy. The results of Figure 7 showed that the expression of the proteins TNFR2, PIPK1, and MLKL was increased with Lu01-M treatment (Figure 7a). The expression of the negative regulator caspase-8 was not affected by Lu01-M (Figure 7b). The results suggested that Lu01-M may induce cell death via necroptosis. To confirm the roles of necroptosis mechanism in Lu01-M-induced anti-proliferation, the necroptosis inhibitor nec-1 was used. The results showed that the combined treatment of Lu01-M and nec-1 enhanced the Lu01-M-induced cell death (Figure 7c).

### 3.8. L01-M-Induced Cell Autophagy Promoting Cell Survival Response in Human Prostate Cancer PC3 Cells

In addition to apoptosis and necroptosis, autophagy also plays an important role in cancer treatment. Autophagy has been known to play a role in cell survival mechanisms and to protect the cell from death. The expression of p62 and LC3B proteins, markers of autophagy, was increased with Lu01-M treatment as observed by confocal microscope (Figure 8a,b). Other autophagy-related proteins were also detected by Western blotting analysis. Lu01-M treatment promoted the expression of p62, LC3 I, and LC 3II proteins and suppressed the expression of p-mTOR (Figure 8c). To clarify the role of autophagy on cell survival following Lu01-M treatment, the autophagy inhibitor 3-MA was used. The inhibitor not only reduced cell viability but also enhanced the cytotoxicity by Lu01-M (Figure 8d). Autophagy was also found to play a role in cell survival mechanisms while the cell is damaged [14]. These results suggested that autophagy played a role in cell survival in Lu01-M-induced anti-proliferation.

## 4. Discussion

Many studies suggested that the marine organism Actinomycetes produces preservatives, insecticides, antibacterial, cytotoxic, and anti-inflammatory agents [15]. In our study, we used physical and chemical methods to isolate marine sediments. We screened the marine actinomycetes *Streptomyces* sp. that exhibited cytotoxic activity and applied fermentation techniques to obtain larger quantities of the extract. The metabolites crude extract Lu01-M was obtained with ethyl acetate as the solvent (Table 1 and Table 2; Figure 1).

MTT assay and colony formation assay demonstrated Lu01-M anti-proliferative activity (Figure 2 and Figure 3). Moreover, apoptosis, cell cycle G2/M phase arrest, MMP disruption, ER stress, DNA damage, necroptosis, and autophagy were observed in prostate cancer cell lines after Lu01-M treatment (Figure 2, Figure 4, Figure 5, Figure 6, Figure 7 and Figure 8). Lu01-M induced apoptosis by promoting PARP cleavage and Sub-G1 level (Figure 2). Both PARP cleavage and sub-G1 are standard markers of apoptosis [16]. The mitochondrial membrane potential was decreased, and cytochrome *c* release was increased into the cytosol (Figure 5a–d). ER stress-related proteins including, PERK, IRE1, Grp78/Bip, p-eIF-2α, ASK-1, and p-JNK were increased after Lu01-M treatment, but the expression of ATF6 was suppressed (Figure 5e). Grp78/Bip is a central regulator of the endoplasmic reticulum. The increase in GRP78/Bip stimulates the survival response while causing ER stress [17]. PERK, eIF-2α, ATF-4, and ATF6 are involved in the CHOP pathway of ER stress [18]. IRE1, ASK-1, and p-JNK play important roles in the ER stress pathway and are linked to the mitochondrial pathway [18]. These results suggested that Lu01-M-induced apoptosis may involve the ER–mitochondrial pathway.

Lu01-M resulted in cell cycle G2/M phase arrest in a dose-dependent manner via the suppression of CDK4, CDK6, p-cdc2, cyclin B1, p-Rb, and E2F1, and the promotion of p21 (Figure 4). p21 (also known as p21^WAF1/Cip1^) plays an important role in regulating cell cycle arrest in response to stimuli [19]. The cyclin B1/CDK1 complex was found to promote cells into mitosis [19]. CDK4/CDK6 and Rb play a role in the G1 phase but also affect mitotic machinery that is regulated by p21 [20]. The results suggested that Lu01-M induced prostate cancer cell lines cell cycle G2/M phase arrest. Interestingly, the G2/M phase is often caused by DNA damage [21]. In this study, DNA damage response was observed (Figure 6). pATM and pChk2 are mainly responsible for DSB signaling, while ATR and Chk1 are the checkpoint kinases activated when ssDNA accumulates. These results confirmed the DNA damage and cycle arrest of Lu01-M against PC3 cell lines.

In addition to apoptosis and cycle arrest, necroptosis also plays an important role in Lu01-M-induced cytotoxic activity. Lu01-M promoted the expression of TNFR2, RIP1, and MLKL with no effect on caspase-8 in PC3 cells (Figure 7a,b). TNFR, RIP1, and MLKL are markers of necroptosis [22]. Additionally, Nec-1, the necroptosis inhibitor targeting RIPK1 protein, was used. However, Nec-1 enhanced Lu01-M induced anti-proliferation (Figure 7c). These results suggested that Lu01-M induced cell necroptosis, but Nec-1 could not reduce Lu01-M-induced cell death. It means that the role of Lu01-M-induced necroptosis was unclear.

Furthermore, the expression of autophagy markers P62, LC3-I, and LC3-II [23] was increased with Lu01-M treatment (Figure 8a–c). mTOR protein, the negative regulator of autophagy, was decreased (Figure 8c). However, the Lu01-M-induced cytotoxicity was enhanced when treated with autophagy inhibitor (3-MA). Autophagy has been known to play an important role in cell survival mechanisms [14]. These indicated that the cell autophagy that Lu01-M induced caused cell survival response.

## 5. Conclusions

In summary, we obtained the metabolite Lu01-M that was isolated from nature marine *Actinomycetes*. Lu01-M inhibited cell proliferation and migration via cell apoptosis, cycle arrest, and DNA damage in the prostate cancer cell line. These results indicated that L01-M has the potential to be developed as an agent against prostate cancer.

## Figures and Tables

**Figure 1 life-11-01414-f001:**
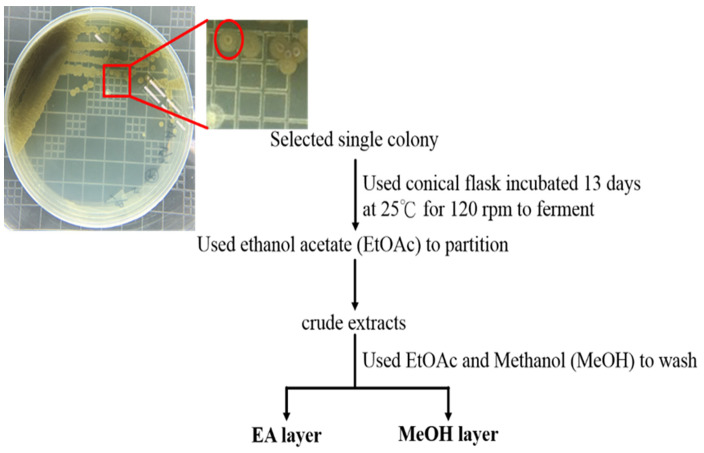
The extracted processes of *Streptomyces* sp. fermentation.

**Figure 2 life-11-01414-f002:**
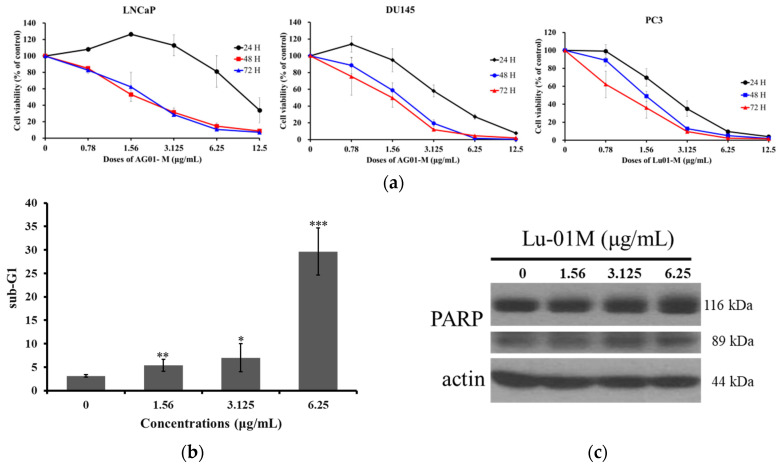
Effect of Lu01-M on cell proliferation in human prostate cancer cells. (**a**) Human prostate cancer cells PC3, DU145, and LNCaP were treated with Lu01-M at different concentrations for 24, 48, and 72 h and were subjected to MTT assay. The results are presented as means ± SD of three independent experiments. (**b**) PC3 cells were treated with the indicated concentration of Lu01-M for 24 h, and sub-G1 was analyzed by flow cytometry. The quantitative results are presented as means ± SD of three independent experiments. * Compared with the control (* *p* < 0.05; ** *p* < 0.01; *** *p* < 0.005). (**c**) PARP, an apoptotic marker, was detected by Western blotting analysis. Actin was used as an internal control to show the equal loading of the proteins.

**Figure 3 life-11-01414-f003:**
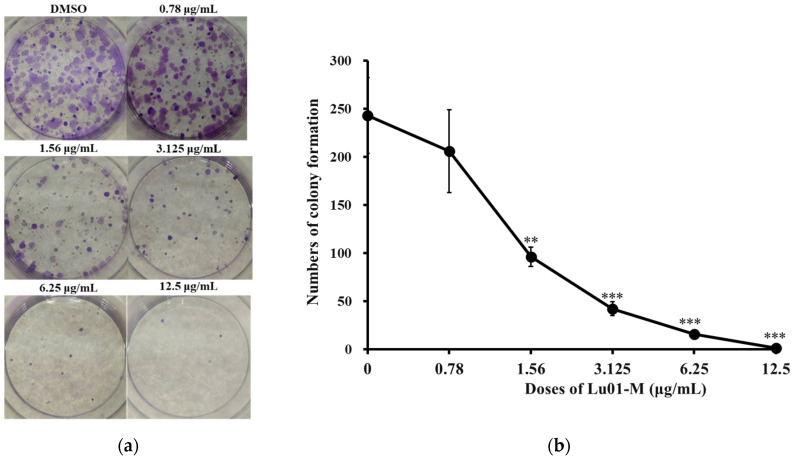
Lu01-M inhibited colony formation and cell migration in human prostate cancer PC3 cells. (**a**) Colony formation assay of PC3 prostate cancer cells. Cells were treated with different concentrations of Lu01-M for 12 days. (**b**) Quantitative results are presented as means ± SD of three independent experiments. (**c**) Wound-healing assay of PC3 cells incubated without or with the different doses of Lu01-M and was evaluated by inverted optical microscopy. (**d**) Quantitative analysis of the relative cell migration was performed after 3, 6, 9, and 12 h. Quantitative results are presented as means ± SD of three independent experiments. * Compare with the control (* *p* < 0.05; ** *p* < 0.01; *** *p* < 0.005).

**Figure 4 life-11-01414-f004:**
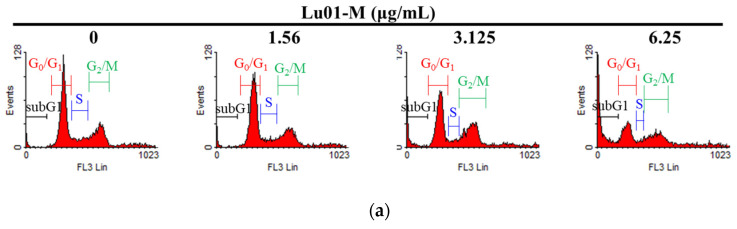
Lu01-M caused cell cycle G2/M phase arrest in PC3 cells. (**a**) Cells were treated with the indicated concentration of Lu01-M for 24 h and were analyzed with flow cytometry. (**b**) Quantitative results are presented as means ± SD of three independent experiments. * Compared with the control (* *p* < 0.05). (**c**) The expression of cell cycle-regulated protein was determined with Western blotting assay. Actin was used as an internal control to show the equal loading of the proteins.

**Figure 5 life-11-01414-f005:**
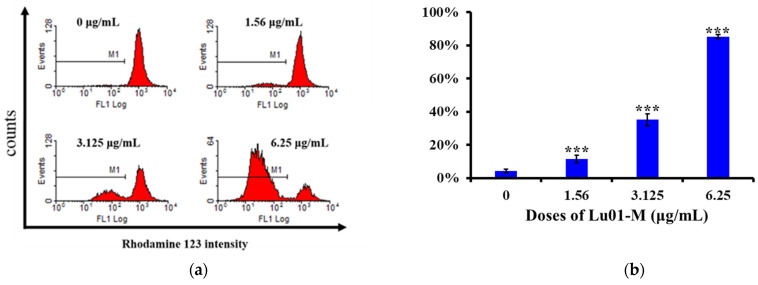
Lu01-M induced MMP disruption and ER stress in PC3 cells. (**a**) Cells were treated with the indicated concentration of Lu01-M for 24 h, and the disruption of MMP was assessed with Rhodamine123 staining using flow cytometric analysis. M1 stands for the percentage of cells with low MMP. (**b**) Quantification of PC3 cells with the disruption of MMP. (**c**) PC3 cells were treated with the indicated concentration of Lu01-M for 24 h, and cytosolic proteins were collected. Cytochrome c expression was detected by Western blotting analysis. (**d**) Quantitative results are presented as means ± SD of three independent experiments. * Compared with the control (* *p* < 0.05; ** *p* < 0.01; *** *p* < 0.005). (**e**) The expression of ER stress-related proteins was determined with a Western blotting assay. Actin was used as an internal control to show the equal loading of the proteins.

**Figure 6 life-11-01414-f006:**
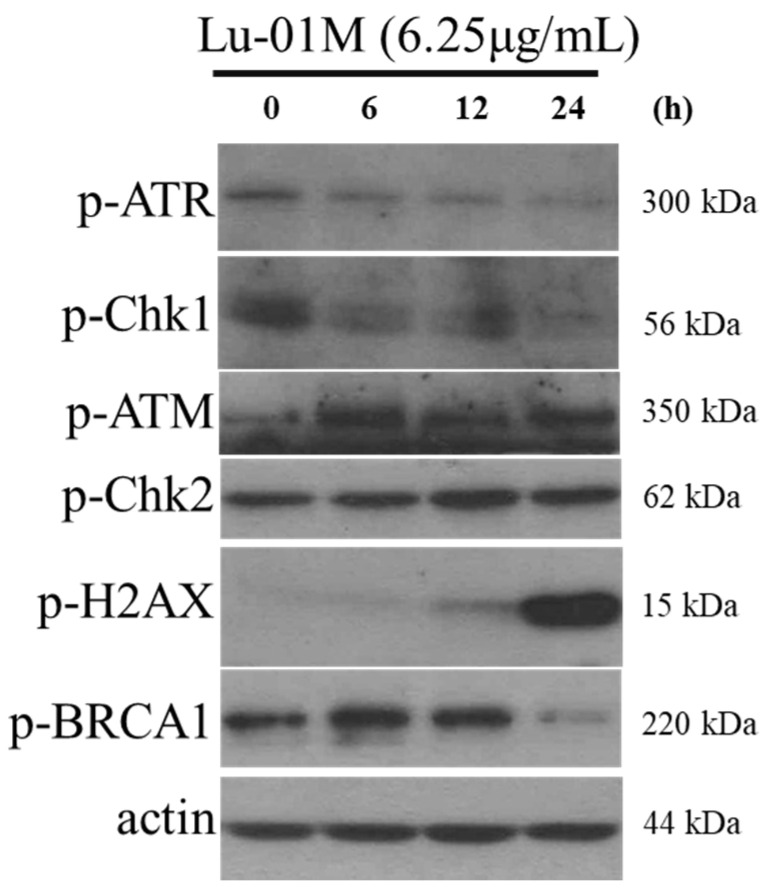
Effect of Lu01-M on the DNA damage of human prostate cancer PC3 cells. Lu01-M induced the phosphorylation of proteins involved in the DNA damage signaling pathway. The biomarkers of DNA damage were determined with a Western blotting assay. PC3 cells were treated with 6.25 μg/mL of Lu01-M for the indicated times. Actin was used as an internal control to show the equal loading of the proteins.

**Figure 7 life-11-01414-f007:**
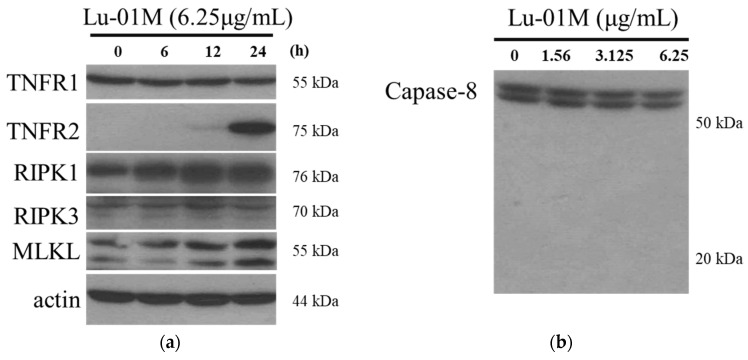
Lu01-M-induced PC3 cells necroptosis. Cells were treated with 6.25 μg/mL of Lu01-M for the indicated times. (**a**) The necroptosis-related proteins were detected by Western blotting analysis. (**b**) Caspase-8 expression was detected by Western blotting analysis. (**c**) Cells were pretreated with or without 19 μM of Nec-1 for 2 h and then were treated with the indicated concentrations of Lu01-M for 24 h with MTT assay. Quantitative results are presented as means ± SD of three independent experiments (* *p* < 0.05; *** *p* < 0.005). Actin was used as an internal control to show the equal loading of the proteins.

**Figure 8 life-11-01414-f008:**
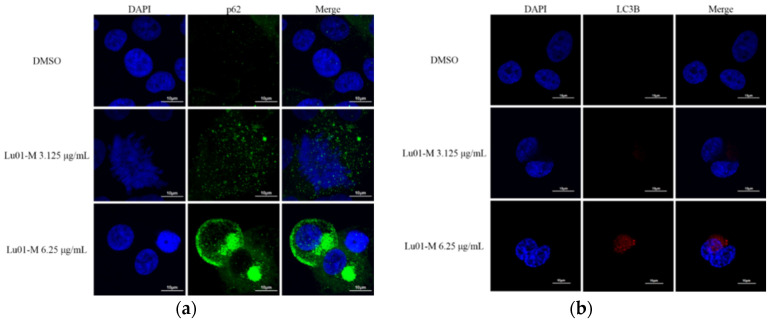
Lu01-M-induced cell autophagy in PC3 cells. Cells were treated with the indicated concentration of Lu01-M (6.25 μg/mL) for 24 h. (**a**,**b**) The effect of Lu01-M on p62 and LC3 proteins as observed by immunofluorescence using confocal microscope. (**c**) Autophagy-related proteins were detected by Western blotting analysis. (**d**) Cells were pretreated with or without 1 mM of 3-MA for 2 h and then were treated with the indicated concentrations of Lu01-M and were subjected to MTT assay for 24 h. Quantitative results are presented as means ± SD of three independent experiments (* *p* < 0.05).

**Table 1 life-11-01414-t001:** Antibacterial activity of bacterial isolates against *S. aureus* and *B. subtilis* growth.

Strain	Activity Against (mm)	
	*Staphylococcus aureus*	*Bacillus subtilis*
**Ac**	-	4.3
**Lu01**	1.3	-
**Ah**	20.3	15.6
**Bi**	-	11.3
**Db**	-	*
**De**	*	-

-: no inhibition zone; *: weak inhibition zone (less than 1 mm).

**Table 2 life-11-01414-t002:** The gene sequence of marine actinomycetes with antibacterial activity, compared with the closest two results from the NCBI database.

Strain	Accession No.	Description	% Identity
**Ac**	NR_041213.1 NR_043490.1	*Streptomyces pulveraceus* *Streptomyces atratus*	99 99
**Lu01**	KM370054.1 KF793806.1	*Streptomyces* sp. *FoRh86* *Streptomyces* sp. *SW4*	99 99
**Ah**	KF287177.1 X87320.1	*Streptomyces* sp. *PN1018* *Streptomycetaceae*	100 99
**Bi**	NR_029370.1 NR_027229.1	*Saccharomonospora* *azurea* *Saccharomonospora glauca*	100 99

## Data Availability

Not applicable.

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
