# Peer review of "The Anti-Proliferative Activity of Secondary Metabolite from the Marine Streptomyces sp. against Prostate Cancer Cells"

_life, 2021, doi:10.3390/life11121414_

Round 1

Reviewer 1 Report

General comments:

The authors isolated a new compound Lu01-M. In general, it is a novel secondary metabolite from the marine actinomycetes Streptomyces sp.. They reported that Lu01-M induced cytotoxic activity through multiple mechanisms including G2/M arrest, apoptosis, necroptosis, autophagy, ER stress, DNA damage, and inhibiting colony formation and cell migration.

Minor comments:
1. For the study of new compounds, the toxicity of normal cells is suggested to be explored. Alternatively, the drug safety of Lu01-M needs to be discussed.

2. The label of G1 should be corrected G0/G1.

3. The author pointed out the S phase of the cell cycle is statistically significant in Figure 4b, but the image is not easy obvious.

4. In Figure 5, the expression of cytochrome c in the cytoplasm is detected, but the extraction method is not described in the method.

5. “3.7. Lu01-M caused DNA damage in human prostate cancer PC3 cells.” The “DNA damage” in section 3.7 needs to correct to “necroptosis”.

6. Please consider unification of the font size and style in your figures. Placement of panel labels is unusual and should be changed.

7. Chk1 and Chk2 are spelled with capital “C”

Author Response

10th Dec, 2021

Dear Assistant Editor Ms. Ma,

    We hereby submit the revised manuscript (Manuscript ID: life-1496399) “The anti-proliferative activity of secondary metabolite from the marine Streptomyces sp. against prostate cancer cells”. Our manuscript has been revised according to the reviewer’s comments using the “Track Changes”. All reviewer’s questions have been answered in details in the following text. We deeply appreciate for your kind help and concern. We will be grateful if the manuscript can be considered for publication in Life.

Please find below our response to the reviewers’ comments.

We hope our response to the reviewers’ comments is satisfactory and meets Life standards. We have tried to support every claim with the corresponding recent references and needed experiments. We also hope that our revised manuscript is acceptable for publication.

Thank you again for your time and concern.

With my best regards

Sincerely yours,

Mei-Chin Lu

Email: jinx6609@nmmba.gov.tw

08-8825001#1354

Graduate Institute of Marine Biology, National Dong Hwa University.

National Museum of Marine Biology & Aquarium

Reply to the Reviewers’ Comments

Manuscript ID: life-1496399

Manuscript Title: The anti-proliferative activity of secondary metabolite from the marine Streptomyces sp. against prostate cancer cells

We would like to express our deep appreciation for your concern regarding the submitted manuscript. We are really thankful for your quick and informative reply. We are also grateful for the valuable comments and suggestions from the reviewers aiming to improve the scientific quality of the submitted manuscript. After reading the reviewers’ comments and discussing them with other coauthors, we have corrected the manuscript accordingly. We have replied to the comments and questions in a point-by-point fashion. Here we enclose the response to the reviewers’ comments.

Responses to the Reviewer’s Comments

General comments:

The authors isolated a new compound Lu01-M. In general, it is a novel secondary metabolite from the marine actinomycetes Streptomyces sp.. They reported that Lu01-M induced cytotoxic activity through multiple mechanisms including G2/M arrest, apoptosis, necroptosis, autophagy, ER stress, DNA damage, and inhibiting colony formation and cell migration.

Reviewer 1:

General comments:

The authors isolated a new compound Lu01-M. In general, it is a novel secondary metabolite from the marine actinomycetes Streptomyces sp.. They reported that Lu01-M induced cytotoxic activity through multiple mechanisms including G2/M arrest, apoptosis, necroptosis, autophagy, ER stress, DNA damage, and inhibiting colony formation and cell migration.

Minor comments:
1. For the study of new compounds, the toxicity of normal cells is suggested to be explored. Alternatively, the safety of Lu01-M needs to be discussed.

  • We are grateful for the comments. Actually, Lu01-M is the crude extract. The chemical profiles and safety will be further evaluated in the future study. Alternatively, the effect of Lu01-M on cytotoxicity of normal CCD-966-SK cells (IC50: 1.56 μg/mL) is less toxic than the effect of clinical anticancer drug, Doxorubicin (IC50: 0.0574 μg/mL) for 72 h treatment with MTT assay. Therefore, the result is added in the Section 3.1.
  1. The label of G1 should be corrected G0/G1.
  • Thanks for the reviewer’s comments. According to the reviewer’s suggestion, we revised the G0 label in Figure 4 of the manuscript to G0/G1.
  1. The author pointed out the S phase of the cell cycle is statistically significant in Figure 4b, but the image is not easy obvious.
  • Thanks for the reviewer’s comments. According to the reviewer’s suggestion, we have updated Figure 4a. Since the S phase value of the control group and Lu01-M group (6.25 μg/mL) is only 3% different, it is not easy to observe from the picture.
  1. In Figure 5, the expression of cytochrome c in the cytoplasm is detected, but the extraction method is not described in the method.
  • Thanks for the reviewer’s comments. Since we used total proteins for analysis and did not separate the cytoplasm from the nucleoprotein, to reduce readers’ misunderstandings, we amended the description of the cytochrome c quantification results (Lines 257-259).
  1. “3.7. Lu01-M caused DNA damage in human prostate cancer PC3 cells.” The “DNA damage” in section 3.7 needs to correct to “necroptosis”.
  • Thanks for the reviewer’s comments. We have corrected the mistake.
  1. Please consider unification of the font size and style in your figures. Placement of panel labels is unusual and should be changed.
  • Thanks for reviewer’s comments. According to the reviewer’s suggestion, we re-check and correct the figures in the manuscript.
  1. Chk1 and Chk2 are spelled with capital “C”

Thanks for reviewer’s comments. According to the reviewer’s suggestion, we changed the spelling of "C" in Chk1 and Chk2 in the article and Figure 6.

Reviewer 2 Report

The authors have carried out quite an interesting and rich work on the study of the biological activity of Lu01-M, a secondary metabolite from the marine actinomycetes Streptomyces sp. This work will be of interest both to researchers directly involved in the study of these objects, and to chemists engaged in the field of medicinal chemistry.
Meanwhile, before giving the go-ahead for publication, the authors should clarify a number of issues that arose during the review of this manuscript.
1. I believe that this article will only benefit if the authors add the chemical structures of the compounds mentioned in the text. In addition, there is no data on the methods of their identification, purity and homogeneity. How was the structure and purity of individual chemical compounds determined?
2. What equipment (brand of cytometer, spectrophotometer, etc.) and software were used in the biological tests? What software was used to process the cell cycle?
3. I would also like the authors to step by step describe the calculation of the error, using at least one experiment as an example, when calculating the IC50, as well as the preparation and titration of stock solutions for their determination using MTT.

Author Response

10
th Dec , 20 21
Dear
Assistant Editor Ms. Ma
We hereby submit the revised manuscript (Manuscript ID: life-1496399) “The anti-proliferative activity of secondary metabolite from the marine Streptomyces sp. against prostate cancer cells”. Our manuscript has been revised according to the reviewer’s comments using the “Track Changes”. All reviewer’s questions have been answered in details in the following text. We deeply appreciate for your kind help and concern. We will be grateful if the manuscript can be considered for publication in Life.
Please find below our response to the reviewers’ comments
We hope our response to the reviewers’ comments is satisfactory and meets
Life standards. We have
tried to support every claim with the corresponding recent references and ne eded experiments. We also hope that our revised manuscript is acceptable for publication.
Thank you again for your time and concern.
With my best regards
Sincerely yours,
Mei
Chin Lu
Email:
jinx6609@nm mba.gov.tw
08
8825001#1354
Graduate Institute of Marine Biology, National Dong Hwa University.
National Museum of Marine Biology & Aquarium
Reply to the Reviewers’ Comments
Manuscript ID: lifeife--14963991496399
Manuscript Title: The anti-proliferative activity of secondary metabolite from the marine Streptomyces sp. against prostate cancer cells
We would like to express our deep appreciation for your concern regarding the submitted manuscript. We are really thankful for your quick and informative reply. We are also grateful for the valuable comments and suggestions from the reviewers aiming to improve the scientific quality of the submitted manuscript. After reading the reviewers’ comments and discussing them with other coauthors, we have corrected the manuscript accordingly. We have replied to the comments and questions in a point-by-point fashion. Here we enclose the response to the reviewers’ comments.
Responses to the Reviewer’s Comments
General comments:
The authors isolated a new compound Lu01-M. In general, it is a novel secondary metabolite from the marine actinomycetes Streptomyces sp.. They reported that Lu01-M induced cytotoxic activity through multiple mechanisms including G2/M arrest, apoptosis, necroptosis, autophagy, ER stress, DNA damage, and inhibiting colony formation and cell migration.
Reviewer 2:
Comments and Suggestions for Authors The authors have carried out quite an interesting and rich work on the study of the biological activity of Lu01-M, a secondary metabolite from the marine actinomycetes Streptomyces sp. This work will be of interest both to researchers directly involved in the study of these objects, and to chemists engaged in the field of medicinal chemistry. Meanwhile, before giving the go-ahead for publication, the authors should clarify a number of issues that arose during the review of this manuscript. 1. I believe that this article will only benefit if the authors add the chemical structures of the compounds mentioned in the text. In addition, there is no data on the methods of their identification,
purity and homogeneity. How was the structure and purity of individual chemical compounds determined? ✓ We appreciate the comments. Lu01-M is the crude extract from marine actinomycetes Streptomyces sp.. The extracted method is described in Section 3.1.. Alternatively, the chemical profiles of Lu01-M will be further evaluated with bioactivity-guided fractionation in the future study. 2. What equipment (brand of cytometer, spectrophotometer, etc.) and software were used in the biological tests? What software was used to process the cell cycle? ✓ Thanks for reviewer’s comments. According to the reviewer’s suggestions, we added the brands of biological testing equipment and software in the materials and methods. 3. I would also like the authors to step by step describe the calculation of the error, using at least one experiment as an example, when calculating the IC50, as well as the preparation and titration of stock solutions for their determination using MTT.
✓ Thanks for reviewer’s commentsThanks for reviewer’s comments. According to the reviewer’s suggestions, we add the relevant . According to the reviewer’s suggestions, we add the relevant description to the materials and method of MTT assay.description to the materials and method of MTT assay.
